biomedical engineering

non-contact sensor, impulse radio ultrawideband radar, newborn, respiratory rate, neonatal intensive care unit

**Authors for correspondence:**
Sung Ho Cho
e-mail: dragon@hanyang.ac.kr
Hyun-Kyung Park
e-mail: neopark@hanyang.ac.kr

†These authors contributed equally to this study.

# Non-contact respiration monitoring using impulse radio ultrawideband radar in neonates

Jong Deok Kim[1,†], Won Hyuk Lee[2,†], Yonggu Lee[4,†], Hyun Ju Lee[3], Teahyen Cha[1], Seung Hyun Kim[1], Ki-Min Song[6], Young-Hyo Lim[4], Seok Hyun Cho[5], Sung Ho Cho[2] and Hyun-Kyung Park[3]

[1]Department of Pediatrics, Hanyang University College of Medicine, Seoul, Republic of Korea
[2]Department of Electronics and Computer Engineering, Hanyang University, 222 Wangsimni-ro, Seongdong-gu, Seoul 04763, Republic of Korea
[3]Division of Neonatology, Department of Pediatrics, Hanyang University College of Medicine, Seoul, Republic of Korea
[4]Division of Cardiology, Department of Internal Medicine, and [5]Department of Otorhinolaryngology, Hanyang University College of Medicine, Seoul, Republic of Korea
[6]Department of Health Sciences, Graduate School, Hanyang University, Seoul, Republic of Korea

JDK, 0000-0002-4266-5655; WHL, 0000-0002-9187-0049;
Y-HL, 0000-0001-7203-7425; SHyC, 0000-0001-8218-5894;
SHoC, 0000-0002-2393-1428; H-KP, 0000-0001-5956-9208

Vital sign monitoring in neonates requires adhesive electrodes, which often damage fragile newborn skin. Because impulse radio ultrawideband (IR-UWB) radar has been reported to recognize chest movement without contact in adult humans, IR-UWB may be used to measure respiratory rates (RRs) in a non-contact fashion. We investigated the feasibility of radar sensors for respiration monitoring in neonates without any respiratory support to compare the accuracy and reliability of radar measurements with those of conventional impedance pneumography measurements. In the neonatal intensive care unit, RRs were measured using radar ($RR_{Rd}$) and impedance pneumography ($RR_{IP}$) simultaneously. The neonatal voluntary movements were measured using the radar sensor and categorized into three levels (low [$M_0$], intermediate [$M_1$] and high [$M_2$]). $RR_{Rd}$ highly agreed with $RR_{IP}$ ($r = 0.90$; intraclass correlation coefficient [ICC] = 0.846 [0.835−0.856]). For the $M_0$ movement, there was good agreement between $RR_{Rd}$ and $RR_{IP}$ (ICC = 0.893; mean bias −0.15 [limits of agreement (LOA) −9.6 to 10.0]). However, the agreement was slightly lower for the $M_1$ (ICC = 0.833; mean bias = 0.95 [LOA −11.4 to 13.3]) and $M_2$

(ICC = 0.749; mean bias = 3.04 [LOA −9.30 to 15.4]) movements than for the $M_0$ movement. In conclusion, IR-UWB radar can provide accurate and reliable estimates of RR in neonates in a non-contact fashion. The performance of radar measurements could be affected by neonate movement.

# 1. Introduction

Respiratory rates (RRs) in neonates in the neonatal intensive care unit (NICU) are most often monitored using impedance pneumography (IP), which measures the changes in electrical impedance resulting from aeration of the lung and movement of the chest wall through electrodes attached to the chest skin. One major limitation of IP is electrode attachment, which often causes damage, including eruptions, abrasions and ulcerations, to fragile neonatal skin [1]. The expendable supplies, such as electrodes or cables, for IP may also serve as sources of contagious diseases and require regular replacement [2,3]. Recently, several novel methods for monitoring respiration without direct contact with the subjects have been introduced [4–7]. However, the feasibility or validity of those methods have seldom been evaluated in actual clinical practices. Moreover, only a few studies have evaluated the performance of non-contact respiration monitors in neonates [8–11].

Radar systems detect object motions by transmitting a band of radio waves and receiving the reflected waves from the object. By occupying a wide frequency band of greater than 500 MHz (25% of the fractional bandwidth), ultrawideband radar systems could provide many advantages, including high resolution, multipath resistance, good penetration power and a simple hardware structure [12,13]. The impulse radio ultrawideband (IR-UWB) radar system uses impulse waves of a short duration. Because of its high resolution and good penetration power, an IR-UWB radar system can recognize subtle movements of human body parts through clothing [14,15]. Previous reports have shown that an IR-UWB radar system could measure the chest movement of clothed human subjects [16–18].

Therefore, when its feasibility and validity can be ensured, the IR-UWB radar system may provide a valuable alternative for respiration monitoring in neonates. The aims of this study are to investigate the feasibility of the IR-UWB radar sensor as a respiration monitor for neonates and to evaluate the accuracy and reliability of the radar sensor by comparison with those of a conventional IP monitor.

# 2. Material and methods

## 2.1. Subjects

The study protocol was approved, and the study processes were monitored by the Institutional Review Board of Hanyang University Hospital, Seoul, Korea (no. 2017-09-046-002). Written informed consent was obtained from the parents of every neonate before their enrolment in the study. We prospectively enrolled clinically stable full-term neonates who were admitted to the NICU from March to April 2018. Neonates who required incubator care or respiratory support or had unstable vital signs, including dyspnea, sustained tachypnea (RR > 60 breaths/min) or fever (greater than 38°C), were excluded because continuous data collection was disturbed by frequent interaction by the clinical staff and the baby. Neonates who had any congenital anomalies were also excluded.

## 2.2. Experimental environment

All experiments were conducted at the bedside in the NICU. The radar chip was covered with a white plastic cap. The radar system was hung at the end of a specially designed arm on a tripod and placed at a distance of 35 cm orthogonal to the chest. The neonates were placed in a supine position inside an open-air crib, and the torso of the neonates was covered with a blanket. The experiment was planned to last as long as the baby was stable. The neonates were not touched or repositioned during the data collection, but clinical workflow always took priority over the recordings. The data collected from the radar were processed and stored in a laptop computer placed nearby (figure 1).

## 2.3. Impedance pneumography measurement

To measure RR using IP, a BSM-6501K patient monitor (Nihon Kohden, Tokyo, Japan) was used. IP records changes in the electrical impedance of the patient's thorax. Three transcutaneous electrodes

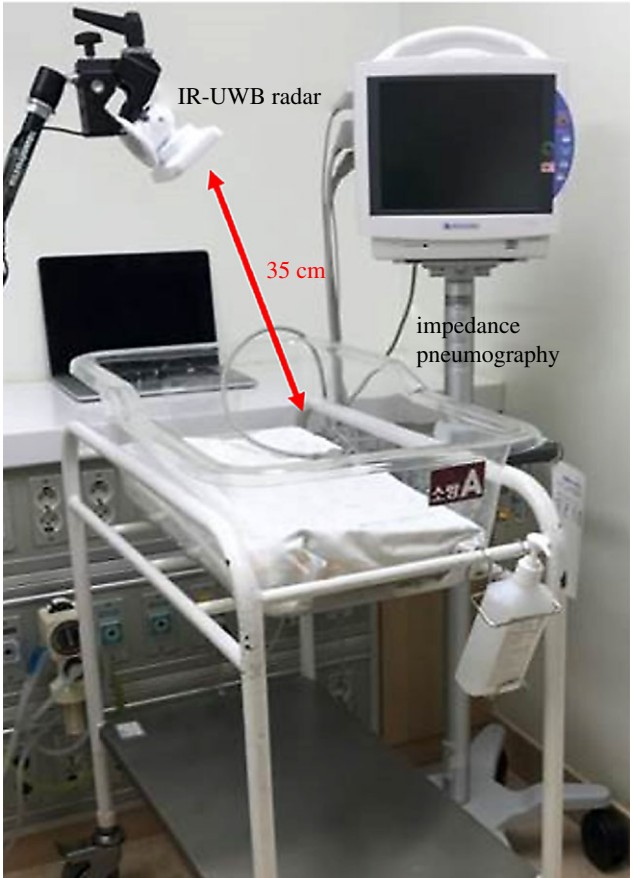

**Figure 1.** Experimental set-up for simultaneous IR-UWB radar and IP recording. A white cap-covered radar sensor (width × depth × height: 5.8 × 3.4 × 1.8 cm; weight: 150 g; inside the actual sensor chip: 2.2 × 1.2 × 0.6 cm, 18 g) is mounted at the end of a specially designed arm. The BSM-6501 K monitor (Nihon Kohden, Tokyo, Japan) is used as a reference.

were attached at the standard positions, and a pulse oximetry sensor was placed on the sole of the neonates. An RR was calculated from the last eight waveforms, recorded on a memory card every second and extracted using viewer software (BSM Viewer, Nihon Kohden, Tokyo, Japan). These RR values were averaged every 10 s for comparison with the radar results.

## 2.4. Radar data collection and processing

A commercially available IR-UWB radar device (X4M06; Xandar Kardian, Delaware, USA) was used to send and collect the radar signals to and from the chest. The IR-UWB radar sensor is capable of detecting objects using the ultrawideband frequency band without interference from other sensors and measuring the distance to the objects. The radar module used has a centre frequency of 7.29 GHz and a bandwidth of 1.5 GHz. It also uses a small power of approximately −41.3 dBm/MHz, which meets Federal Communications Commission standards [19]. MATLAB (MathWorks, New York, MA, USA), which is a commercially available software package, was used for acquiring, processing and storing the data.

Signal processing was automatically performed by a software algorithm [15–17,19], and RR activity was isolated from the heartbeat signal by an algorithm described in our previous study [16,18]. A Fourier transform decomposed a signal with respect to time into a frequency component (figure 2) [16,18]. This was used to extract the frequency component of the RR from the vital sign signal. Figure 3 shows how the frame received by radar is processed as the algorithm progresses for each step in the diagram. The raw data are an unprocessed frame received from the radar, and the clutter removal signal is extracted by removing unwanted background signals received by the radar. In frames received dozens of times per second, the position of the infant can be extracted along the time axis to find a signal containing the respiratory wave. From the extracted breathing signals, the respiration rate can be obtained by going through a filter corresponding to the respiratory frequency band of the neonates. The FFT point size

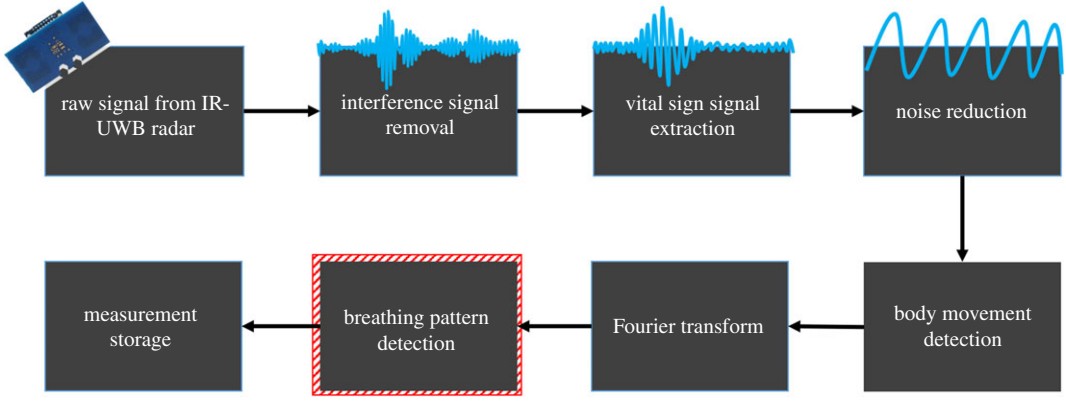

**Figure 2.** Block diagram of the IR-UWB radar signal processing flow for extraction of respiratory signals. Not only the signal from the target (newborn baby) to be observed, but also other surrounding background signals are received every second. Primarily, the unwanted clutter signal must be removed from the raw signal, and the next step is to combine those waveforms, resulting in a better frequency resolution for the vital sign. Because this extracted signal also includes breathing, heartbeat, motion and noise, the measurement noise is reduced by applying the Kalman filter estimation to the time-varying signal, and the range of the radar sensor is configurable through parameter frame stitches. The degree of motion from the extracted signal is integrated with the respiratory waveform; then, the fast Fourier transform algorithm is applied to the constructed vital signal to find the respiratory signal frequency band.

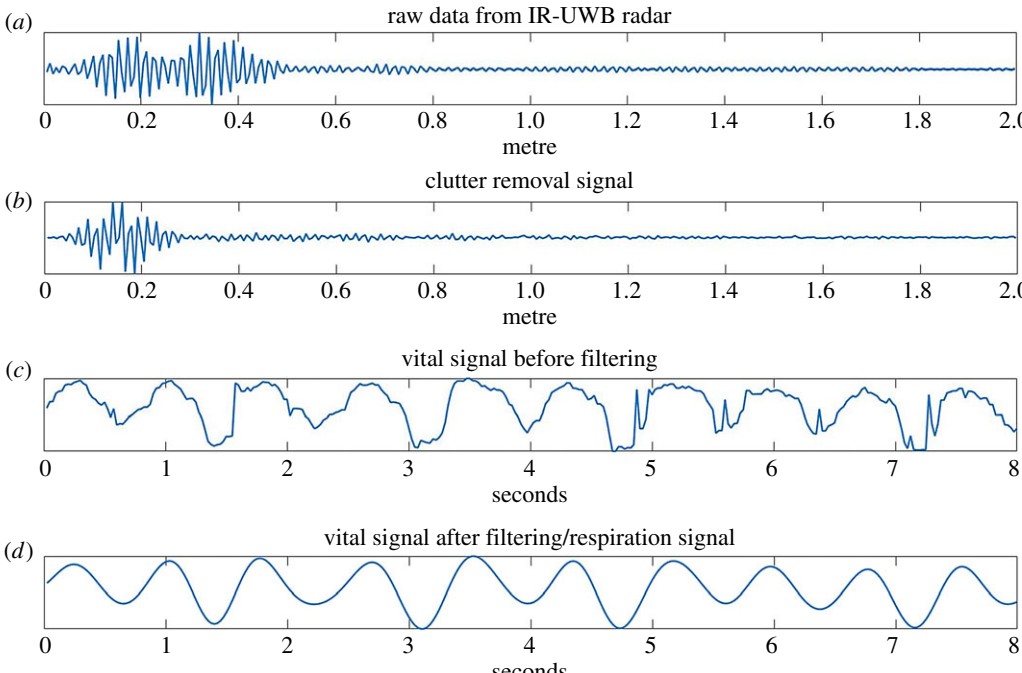

**Figure 3.** Radar signal obtained from each step of the block diagram. (*a*) Raw frame received by radar antenna. (*b*) Clutter removal frame obtained from background subtraction. (*c*) Neonatal respiratory waveform acquired for 8 s. (*d*) Respiratory waveform obtained through low-pass filtering of neonatal respiration frequency.

used is 4096, and the frame per second received by radar is 40. The time-window size used for frequency domain analysis is 8 s.

Respiratory signals were recorded continuously. During experiments, moving extensively, receiving nursing care, repetitive myoclonus, hiccupping, flopping and crying significantly interfered with respiratory signal acquisition. For the final analysis, the RR estimates were averaged on a continuous basis and updated every 10 s, independent of the signal stability, except with the above-mentioned active movements because these estimates allow us to provide comparative data under usual clinical conditions.

## 2.5. Voluntary body movement measurement and classification

To investigate the artefact and interference from voluntary body movement during measurement, the level of movement was integrated based on the change in the distance between the body and the radar system. The radar system transmitted dozens of these signals (frames) to the computer every second [17]. The difference between two frames (previous–current) was calculated (figure 2).

$$m(t) = \sum_{k=1}^{L} |X(t,k) - X(t-1,k)|$$

$$M(t) = \begin{cases} M_0 \ (m(t) < \text{thres}\,1) \\ M_1 \ (\text{thres}\,1 < m(t) < \text{thres}\,2), \\ M_2 \ (m(t) > \text{thres}\,2) \end{cases}$$

where $X(t,k)$ is each frame derived from the radar, $t$ is the time index and $k$ is the distance index. $L$ is the maximum distance reachable by radar. That is, $X(t,k)$ is the magnitude of the signal at each distance at the $t$th time index. The movement value ($m(t)$) can be obtained by adding all the differences between the current frame ($X(t,k)$) and the previous frame ($X(t-1,k)$). Depending on the degree of this movement, the movement value was classified as $M_0$, $M_1$ and $M_2$ via two thresholds: $M_0$, no or slight movements; $M_1$, small movements of the face and extremities; and $M_2$, larger and significant movements/twitches of the extremities and trunk (figure 2). These two thresholds (thres 1 and thres 2) were used to classify the movement levels of the neonates. The thres 1 ($M(t) = 0.06$) and thres 2 ($M(t) = 0.075$) values were defined as the $M(t)$ levels equivalent to hiccupping and crying, respectively.

## 2.6. Statistical analysis

The agreements between $RR_{Rd}$ and $RR_{IP}$ were evaluated using intraclass correlation coefficient (ICC), concordance correlation coefficient (CCC) and Bland–Altman plots with the 2.5% and 97.5% limits of agreement (LOAs). Because our data had multiple measurements in pairs in each patient, we employed one-way ANOVA to calculate the standard deviations of the biases and the LOAs, as previously described [20]. The existence of biases between $RR_{Rd}$ and $RR_{IP}$ was evaluated using a single sample $t$-test. The values of CCCs and mean biases were plotted according to the movement levels ($M_0$, $M_1$ and $M_2$) and the mean RRs (greater than or equal to 40 breaths/min or not). CCCs were compared using the 95% confidence intervals of the estimates, and the biases between $RR_{IP}$ and $RR_{Rd}$ were evaluated using one-way ANOVA with a Tukey HSD test as the post hoc analysis among the groups based on the movements and the mean RRs. All statistical analyses were performed using the statistical software R version 3.4.0 and its packages ICC, epiR and MASS. A value of $p < 0.05$ was considered statistically significant.

# 3. Results

Among 42 full-term neonates (greater than or equal to 37 weeks of gestation) who were admitted to the NICU, six neonates (two males and four females) with a median gestational age of 38 weeks and a median birth weight of 3100 g were included in the study. The median Apgar scores of the neonates were seven at 1 min and nine at 5 min, and the median duration of hospital stay was 12 days. All measurements for the RRs from IR-UWB radar and IP were obtained between 2 and 29 days (median 9 days) after birth (table 1).

The radar recordings were made within the unmodified environment of the NICU. The respiratory signal waveforms from IR-UWB radar appeared to correspond cycle by cycle to the IP waveform. When the intervals between breathing cycles were irregular and even an apnea cycle was presented, the crests and troughs of the respiratory signals from the IR-UWB radar also appeared to be well matched to the IP respiratory signals (figure 4a). Through synchronization of the two data streams, a representative case of measured RRs from both devices in a 5 min period is displayed in figure 4b. The absolute values and the change patterns of RRs are similar between the two devices; when the neonates moved, the discrepancies between the two RRs appeared to increase.

The average recording duration of both the IR-UWB and IP was 134.3 min (range 100–160 min). The data that were obtained when the electrodes were detached from the neonates or when the neonates required feeding, diaper changes or any medical attentions (36% of the total measurement duration) were excluded from the analysis because $RR_{IP}$ was not recognized during these times. RRs were

**Table 1.** Subjects' characteristics. Data are presented as $N$ (%) or the mean (minimum–maximum).

| demographics | $n = 6$ |
| --- | --- |
| gestational age (weeks) | 38.0 (37.0–41.0) |
| birth weight (g) | 3100 (2790–3960) |
| male | 2 (33.3) |
| singleton | 6 (100) |
| small-for-gestational-age (SGA) infant | 1 (16.6) |
| birth by caesarean section | 3 (50) |
| Apgar 1 min | 7 (2–9) |
| Apgar 5 min | 9 (7–10) |
| duration of hospitalization (days) | 12 (5–31) |
| breast milk feeding during hospital stay | 3 (50) |
| age at measurement (days) | 9 (2–29) |
| weight at measurement (g) | 2940 (2680–4060) |

estimated every 10 s in both measurement methods. Consequently, a total of 2974 RRs (1425 for $M_0$, 744 for $M_1$ and 805 for $M_2$) were measured for an average of 85.9 min (range 61–116 min) and were used in the statistical analyses. The RRs measured using IR-UWB radar ($RR_{Rd}$) were highly correlated with the RRs measured using IP ($RR_{IP}$), and the ICC value indicated good agreement between $RR_{Rd}$ and $RR_{IP}$. Although the Bland–Altman plot showed a statistically significant difference between $RR_{Rd}$ and $RR_{IP}$, the absolute value of the mean bias was only 1.17 breaths per minute (bpm) (95% LOA, −10.4 to 12.7 bpm, figure 5a). The numbers of samples, mean RRs, agreement levels and 95% LOAs measured in each patient are summarized in electronic supplementary material, table S1. $M_0$, ICC and CCC showed good agreement between $RR_{Rd}$ and $RR_{IP}$. A Bland–Altman plot showed that the mean bias between $RR_{Rd}$ and $RR_{IP}$ was only 0.21 bpm (95% CI of the mean bias, −0.05 to 0.47), which was not significant, and the width of LOA was also smallest in $M_0$ (table 2 and figure 5b). In $M_1$ and $M_2$, ICCs and CCCs between the two measurements were slightly lower than those in $M_0$, but still high. However, the mean bias levels were significant in both $M_1$ and $M_2$ but not in $M_0$, and the widths of LOA increased as the movement levels increased (table 2 and figure 5c,d). The agreement levels were lower in patients with low mean RRs (less than 40 breaths/min) than those in patients with high mean RRs (greater than or equal to 40 breaths/min; table 2 and figure 5).

As the movement levels increased, CCCs decreased and the mean bias levels increased in patients with high mean RRs. CCCs were not different according to the movement levels, whereas the mean bias level was only slightly higher in $M_2$ than in $M_1$ in patients with low mean RRs. The agreement between $RR_{Rd}$ and $RR_{IP}$ was better in patients with high mean RRs than in those with low mean RR in $M_0$ and $M_1$. However, in $M_2$, the mean bias levels were not different between the patients with high and low mean RRs, and CCC was even worse in the patients with high mean RRs than in those with low mean RRs (figure 6).

## 4. Discussion

To date, various non-contact respiration monitoring methods have been introduced for clinical usage, including camera-based photoplethysmography, continuous wave Doppler radar, infrared thermography, ultrasound, piezoelectric sensors and mattress-integrated capacitors [21,22]. A non-contact respiration monitor would be particularly useful in the field of neonatal care, given that neonates have very fragile skin that could be seriously injured by the attachment and detachment of electrodes for conventional IP monitors [23,24]. Lund et al. [25] have also reported that a single application of adhesive plastic tapes or pectin barriers could disrupt the barrier function of the skin in neonates even without actual skin injuries. Only a few of the above-mentioned non-contact respiration monitoring methods have been evaluated in neonates. Jorge et al. [26] investigated 30 neonates and reported that camera-based photoplethysmography could be used for non-contact continuous monitoring of RR, heart rate and $O_2$ saturation in the NICU. The RR monitor using real-time infrared thermography has also been evaluated in neonates [27]. However, the statistics indicating the extent of agreement between the non-contact

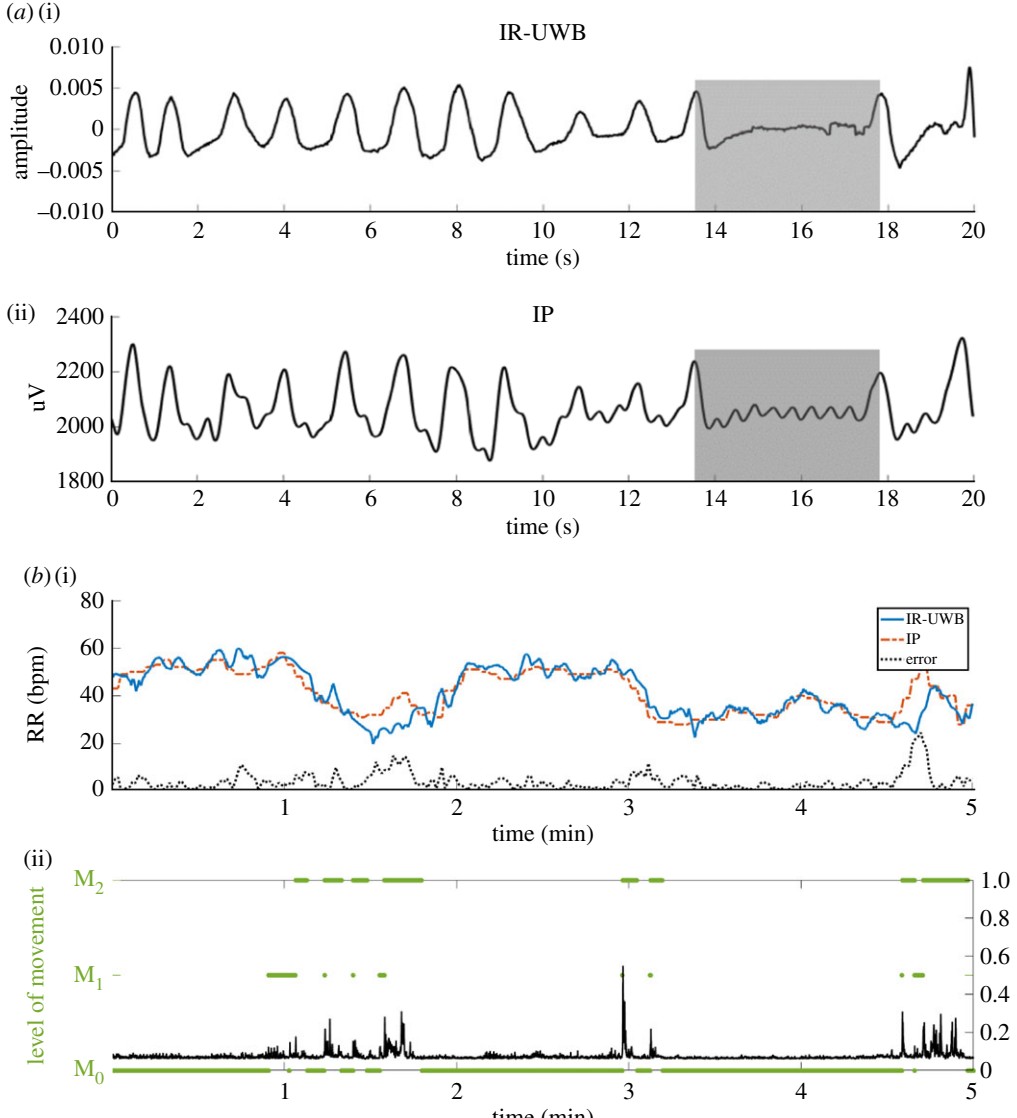

**Figure 4.** Representative results of the raw signal waveform and estimated RRs from both devices. (*a*) Relatively regular breathing signal waveforms from the IR-UWB radar (i) and IP (ii). Short periods of breathing cessation (shaded box), such as apnea or periodic breathing, were comparably detected by IR-UWB radar and IP. (*b*) Measured RRs from the IR-UWB radar system and IP in a 5 min section ($M_0$-dominant). Panels (i) show RR estimates derived from the IR-UWB radar system in blue and the IP-derived estimates in red. Panels (ii) show the level of movement calculated by the IR-UWB radar system. Real movement traces are shown by black waves, and the movements classified into levels are shown with green bars. $M_0$, the segment of no body movement; $M_1$, small movements; $M_2$, larger movements; IP, impedance pneumography; bpm, breaths per minute.

methods and conventional vital sign monitoring methods were not provided in either study. Accordingly, the development of practical non-contact vital sign monitors is highly desired in neonatal care.

By transmitting and receiving a band of radio waves, the IR-UWB radar system can recognize the patient's chest movements during breathing at a distance. Due to its ultrawide bandwidth, the radar can penetrate the patient's clothing or blankets and extract the respiratory signals with high resolution. Because radar is not affected by the status of ambient lights or by the patient's skin colour, it can be used in a nursery even at night when the lights should be dim or off. The measurement system can be located at a more flexible range of distances from the subject with the radar method than with other non-contact vital sign monitoring methods. Compared to other devices, IR-UWB radar sensors typically have small sizes and simple structures, which would provide advantages in installation and operation of such devices. The radio waves used in the IR-UWB radar system have much less power than those in a common Wi-Fi router [16,28], and the radar sensor has been approved by the National Radio Research Agency (Ministry of Science, ICT and Future Planning, Korea; certification no. MSIP-CRM-Top-TSR-M200W) as a safe Wi-Fi device.

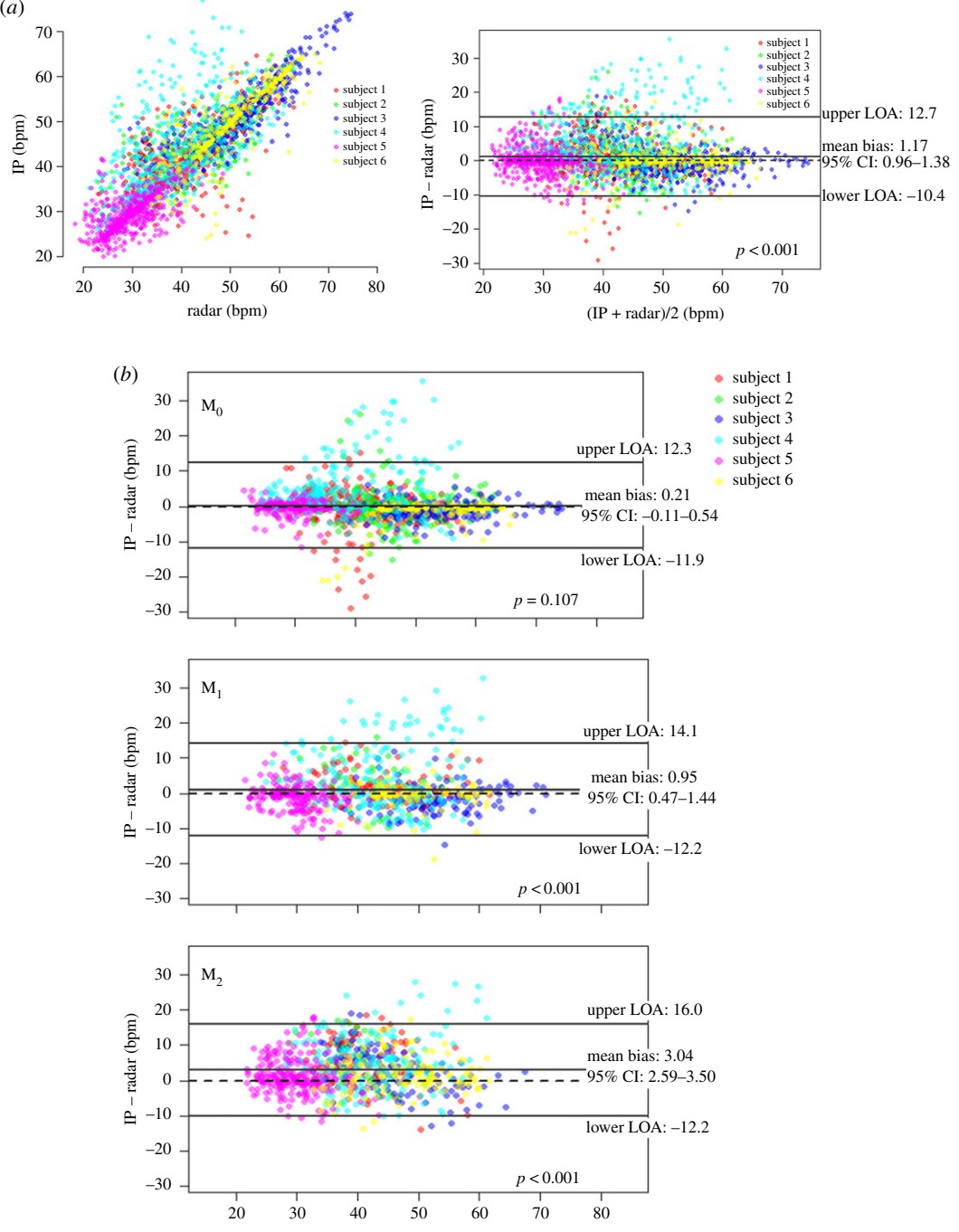

**Figure 5.** Comparison of RR (bpm) measured with the IR-UWB radar system and IP in all six neonates. (a) Scatterplot and Bland–Altman plot between the IR-UWB radar measurements and the IP measurements. A total of 2974 samples were collected from six subjects (average 495.7 ± 137.4 bpm). The mean bias was 1.17 bpm and significant in a one-sample $t$-test. The LOA was −10.4 to 12.7 bpm. (b) Bland–Altman plots between the IR-UWB radar measurements and the IP measurements according to the movement levels. The mean bias levels and the widths of LOA increased significantly as the movement levels increased. The mean bias was not significant in $M_0$, but in $M_1$ and $M_2$, the mean biases were significant and approximately 1.0 and 3.0 bpm, respectively.

Our results have shown that IR-UWB radar could accurately measure RRs in neonates in a non-contact manner. Although movements affected the agreement between the radar and IP measurements, the accuracy was still sufficiently high, even for high levels of movement. The radar was also easily applicable in the NICU environment, where the various surrounding instruments might interfere with the measurements and where the device itself could be an obstacle against daily neonatal care by physicians and nurses. Since we demonstrated the plausibility of the IR-UWB radar as a continuous non-contact vital sign measurement tool in 2017 [17], we have continuously improved

**Table 2.** Agreement between $RR_{Rd}$ and $RR_{IP}$ according to the movements and mean RRs. $RR_{Rd}$, respiratory rate measured in the radar; $RR_{IP}$, respiratory rate measured in the impedance pneumography; RR, respiratory rate; ICC, intraclass correlation coefficient; CCC, concordance correlation coefficient; LOA, limits of agreement; CI, confidence intervals.

| agreement parameter | movement | | | | mean RR of a subject | | |
|---|---|---|---|---|---|---|---|
| | total data | $M_0$ | $M_1$ | $M_2$ | less than 40 min$^{-1}$ (subject 4,5) | $\geq$40 min$^{-1}$ (subject 1,2,3,6) |
| number of cases | 2974 | 425 | 744 | 805 | 1284 | 1690 |
| ICC (95% CI) | 0.846 (0.835−0.856) | 0.893 (0.882−0.903) | 0.833 (0.809−0.853) | 0.749 (0.717−0.778) | 0.721 (0.694−0.746) | 0.831 (0.816−0.846) |
| CCC (95% CI) | 0.846 (0.836−0.856) | 0.893 (0.882−0.903) | 0.833 (0.809−0.853) | 0.754 (0.725−0.782) | 0.725 (0.699−0.748) | 0.831 (0.816−0.845) |
| mean bias (95% CI) | 1.17 (0.96−1.38) | 0.214 (−0.106 to 0.535) | 0.954 (0.473−1.435) | 3.044 (2.587−3.503) | 2.203 (1.840−2.567) | 0.377 (0.084−0.670) |
| LOA | (−10.361), 12.392 | (−11.907), 12.335 | (−12.179), 14.087 | (−12.179), 16.039 | (−10.802), 15.209 | (−11.666), 12.420 |

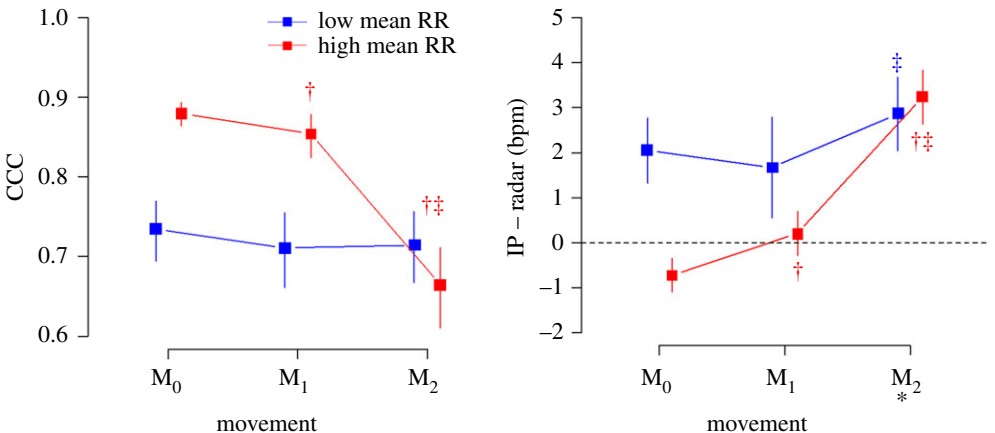

**Figure 6.** Concordance correlation coefficients (CCCs) and the mean bias levels according to the movement levels and mean RRs. The levels of agreement were better in patients with high mean RRs than those in patients with low mean RRs in $M_0$ and $M_1$, whereas the levels of agreement were similar or even worse in patients with high mean RRs compared with those with low mean RRs in $M_2$. $*$ $p > 0.05$ in low versus high mean RRs ($p < 0.05$ in all the other pairs). † $p < 0.05$ versus $M_0$. ‡ $p < 0.05$ versus $M_1$ within the same RR level.

the measurement methodology and signal analysis algorithms. Recently, we reported that heart rates could be accurately monitored using radar in healthy adults and patients with atrial fibrillation who held their breath and cooperated during the measurements [29]. This study is a pilot investigation of the feasibility of radar as an alternative method to conventional contact respiration monitors.

Since the radar system recognizes the movements of a subject, it was more difficult to measure the vital signs accurately using the radar in neonates, from whom no cooperation is expected, than in cooperative adults. Moreover, neonate chest sizes are smaller, and their RRs are more rapid and more variable than those of adults. Through modification of the data processing algorithm, we achieved high levels of accuracy without requiring any constraints during the measurements. Radar signals reflect the diaphragmatic activities and chest wall kinetics, and radar waveforms also contain information regarding the irregularity of respiration (dyspnea), apnea and the amount of movement, as shown in the results. Using this information, we also think that the radar system could be used in complicated clinical situations with mechanical ventilation, the classification of apnea associated with prematurity, the detection of neonatal seizure and sleep–wake cycles.

Our study has several limitations. First, because in principle, the radar system recognizes movements of the subjects, the accuracy of the radar measurements should be affected by vigorous movements. However, most conventional vital sign measurement methods, including IP, are also significantly affected by vigorous movements [21], and the IR-UWB radar system still achieved a good accuracy level, even in $M_2$. There is a short segment of discrepancies at 52–54 min, as shown in figure 3b. Interestingly, a closer look through the video camera could verify a signal perturbation in IP in contrast to the stable radar signal. Secondly, we included only a small number of neonates in stable conditions without any significant birth defects and whose RRs were not too rapid or labile, which may have contributed to the accuracy of the radar measurements. Thirdly, all measurements were obtained in a supine position with the radar sensor set at a fixed angle and distance from the chest. Therefore, the accuracy may be slightly different in other measurement settings.

## 5. Conclusion

The IR-UWB radar system accurately measured neonatal RRs in the NICU environment in a non-contact manner. Although the levels of accuracy could be affected by the movement levels, IR-UWB radar may be a promising option for respiratory monitoring as an alternative to conventional methods, including IP. Future studies will be able to expand the utility of radar for heart rate estimation, apnea detection and movement-based sleep–wake detection.

Ethics. The study protocol was approved and the study processes were monitored by the Institutional Review Board of Hanyang University Hospital, Seoul, Korea (no. 2017-09-046-002).

Data accessibility. The dataset supporting this article has been deposited at the Dryad Digital Repository at: https://doi.org/10.5061/dryad.f3bn03g [30].

Authors' contributions. H.-K.P. and S.Ho.C. conceived the basic ideas, designed the experiments and supervised the processes; W.H.L. and J.D.K. conducted the experiments; J.D.K., W.H.L. and Y.L. conducted data analysis and wrote the manuscript; and T.C., S.H.K., K.-M.S., Y.-H.L. and S.Hy.C. drafted the manuscript. All authors read and approved the final manuscript.

Competing interests. The authors have no competing interests and no conflicts of interest to declare.

Funding. This research was supported by the Bio and Medical Technology Development Program (Next Generation Biotechnology) through the National Research Foundation of Korea (NRF) funded by the Ministry of Science, ICT and Future Planning (grant no. NRF-2017M3A9E2064735).

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
