## [Reviewer comments · Royal Society Open Science]

Review History

RSOS-190149.R0 (Original submission)

Review form: Reviewer 1

Is the manuscript scientifically sound in its present form?

Yes

Are the interpretations and conclusions justified by the results?

Yes

Is the language acceptable?

Yes

Is it clear how to access all supporting data?

Yes

Do you have any ethical concerns with this paper?

No

Have you any concerns about statistical analyses in this paper?

No

Recommendation?

Major revision is needed (please make suggestions in comments)

Comments to the Author(s)

The paper describes some experiments about the detection of the respiration rate in neonates using an IR-UWB RADAR. The work is interesting, but I have some comments to improve the quality of the work:

1. More details about the IR-radar must be included in the text. The only reference to the radar is the company that manufacture it (Xethru X4M06; Xandar Kardian, Delaware, USA).
2. In the formula of the movement $M(t)$, the difference must be defined including an absolute value of the difference $\text{abs}(m(t))$.
3. The signal processing described in fig.2 must be described better. For example, the interference signal removal or vital sign extraction or noise reduction. The detection of RR must be described better. For example, if a FFT is used, what is the time-window period used, the windows used or if another algorithm is used (periodogram, ...).

In order to understand better the procedure, a new figure could be included showing a sample raw signal and the signal after each step (interference reduction, noise reduction and Fourier transform).

4. In order to understand the effect the movement, the Fourier transform for a period with and without movement could be included. It is expected that the signal-to-noise ratios will be worst during movements.

5. The proposed method have two main drawbacks: random movements of the body and the orientation of the body. In the case of neonates the movement is small and the body falls in the illumination area of the body, therefore it is expected that is not a big problem compared with other applications, for example, a person walking. This drawback is studied in the work. The radar detects changes in the chest movement. Therefore, the signal detected in the back or lateral orientation is very small and the signal-to-noise ratio increases. In these situations, the error could be high. Some tests with different orientations of the body with respect to the radar (in lateral orientation or by back) should be included. A solution with two antennas is proposed in [18] to solve the problem. Some additional comments should be included in the discussion about this problem or potential solutions. The results given are only for the most favorable orientation. Some additional results should be included for other body orientations.

Review form: Reviewer 2 (Kim van Loon)**Is the manuscript scientifically sound in its present form?**

Yes

Are the interpretations and conclusions justified by the results?

No

Is the language acceptable?

Yes

Is it clear how to access all supporting data?

Yes

Do you have any ethical concerns with this paper?

No

Have you any concerns about statistical analyses in this paper?

Yes

Recommendation?

Major revision is needed (please make suggestions in comments)

Comments to the Author(s)

Manuscript: Noncontact Respiration Monitoring Using Impulse Radio Ultrawideband (IR-UWB) Radar in Neonates

P1L37 "Neonatal movements" -> Authors probably mean "neonatal voluntary movements" instead of "neonatal chest movements"

P1 Please add information in the abstract about the respiratory status of the PICU neonates: e.g. non-mechanically assisted spontaneous breathing neonates?

P1L39 The authors describe their finding as "excellent agreement" for M0 measurements. But what do define as excellent, because the LOA are rather wide [-9.6-10.0] for a vital parameter with normal values between 30-60 breaths per minute for a neonate.

P2L33 This should be considered as part of your results: "Six neonates were eligible for the experiments among 42 full-term neonates (≥ 37 weeks of gestational age)" and is not a description of the study plan.

P2L36 Why would you exclude neonates with "dyspnea, sustained tachypnea (RR > 60 breaths/min)"? These are the interesting ones, on P2L7 the authors describe that these devices are seldom evaluated in actual clinical practices and I read this as that they are aiming to evaluate IR-UWB in a clinical setting. A device should also perform well in abnormal clinical situations.

P2L40 Did the authors preplanned how long the experiment would take per neonate? Did the authors predefine what they would consider acceptable limits of agreement?

P3L9 Is the averaging time also 10 seconds?

P3L14 I like the way the authors were able to define the extent of movement. I hope that the authors can reassure us that this method does not classify a deep breath as an M1. After all, motion of breathing is not constant, with regular changes between shallow breathing and deep breaths.

P3L37 For this type of data, with repeated measurements for every participant, it is unusual and misleading to present correlation coefficients. This very nicely explained by Bland and Altman. They also wrote an article about methods comparison studies with multiple observations per individual and how this data should be handled differently.

<http://dx.doi.org/10.1080/10543400701329422>

P3L59 I agree with the authors that this breathing trace represents matched respiratory signals. However, this does not explain why, even if only M0 measurements pairs are selected, there is still some considerable spread around the bias (represented by the rather wide LOA), especially at an average breathing rate of 40 breaths/minute. Have the authors looked into detail to these traces to understand why there's such a large discrepancy between the devices in other measurements? Consider for example cardiac rhythm disturbances that interfere with IP signals?

If the authors suspect that a large proportion of the measurement error is due to "failure" of IP measurements it would be worthwhile to look the editorial written by Myles in the Br J Anaesthesia 2007.

P4L10 I would suggest removing the correlation coefficients from the paper and also the correlation plots in Figure 4. Because correlation in this type of research is misleading.

P4L54 I am not fully convinced by the results that IR-UWB can accurately measure respiratory rate in neonates. The bias measurement is excellent, however the LOA are rather wide also when M1 and M2 measurements are removed. A 10 breaths/minute difference can be clinically relevant. Personally, I am not so disturbed by artifacts due to voluntary movement, this indicates that the neonate is active and thus alive. I would suggest that the authors think of clinical situations in which the IR-UWB would detect breathing but the neonate is at risk of respiratory failure. E.g. upper airway obstruction, or epileptic activity. These situations should be recognized and should not produce falsely reassuring respiratory rate numbers.

Figure 5. I do not see the additive value of this figure.

Decision letter (RSOS-190149.R0)

15-Mar-2019

Dear Professor Park,

The editors assigned to your paper ("Noncontact Respiration Monitoring Using Impulse Radio Ultrawideband (IR-UWB) Radar in Neonates") have now received comments from reviewers. We would like you to revise your paper in accordance with the referee and Associate Editor suggestions which can be found below (not including confidential reports to the Editor). Please note this decision does not guarantee eventual acceptance.

Please submit a copy of your revised paper before 07-Apr-2019. Please note that the revision deadline will expire at 00.00am on this date. If we do not hear from you within this time then it will be assumed that the paper has been withdrawn. In exceptional circumstances, extensions may be possible if agreed with the Editorial Office in advance. We do not allow multiple rounds of revision so we urge you to make every effort to fully address all of the comments at this stage. If deemed necessary by the Editors, your manuscript will be sent back to one or more of the original reviewers for assessment. If the original reviewers are not available, we may invite new reviewers.

- Data accessibility

If you wish to submit your supporting data or code to Dryad (<http://datadryad.org/>), or modify your current submission to dryad, please use the following link:
<http://datadryad.org/submit?journalID=RSOS&manu=RSOS-190149>

- Competing interests

- Authors' contributions

- Acknowledgements

- Funding statement

on behalf of Dr Derek Abbott (Associate Editor) and Professor R. Kerry Rowe (Subject Editor)
openscience@royalsociety.org

Comments to Author:

Reviewers' Comments to Author:

Reviewer: 1

Comments to the Author(s)

The paper describes some experiments about the detection of the respiration rate in neonates using an IR-UWB RADAR. The work is interesting, but I have some comments to improve the quality of the work:

1. More details about the IR-radar must be included in the text. The only reference to the radar is the company that manufactures it (Xethru X4M06; Xandar Kardian, Delaware, USA).

2. In the formula of the movement $M(t)$, the difference must be defined including an absolute value of the difference $\text{abs}(m(t))$.

3. The signal processing described in fig.2 must be described better. For example, the interference signal removal or vital sign extraction or noise reduction. The detection of RR must be described better. For example, if a FFT is used, what is the time-window period used, the windows used or if another algorithm is used (periodogram, ...).

In order to understand better the procedure, a new figure could be included showing a sample raw signal and the signal after each step (interference reduction, noise reduction and Fourier transform).

4. In order to understand the effect the movement, the Fourier transform for a period with and without movement could be included. It is expected that the signal-to-noise ratios will be worst during movements.

5. The proposed method has two main drawbacks: random movements of the body and the orientation of the body. In the case of neonates the movement is small and the body falls in the illumination area of the body, therefore it is expected that is not a big problem compared with other applications, for example, a person walking. This drawback is studied in the work. The radar detects changes in the chest movement. Therefore, the signal detected in the back or lateral orientation is very small and the signal-to-noise ratio increases. In these situations, the error could be high. Some tests with different orientations of the body with respect to the radar (in lateral orientation or by back) should be included. A solution with two antennas is proposed in [18] to solve the problem. Some additional comments should be included in the discussion about this problem or potential solutions. The results given are only for the most favorable orientation. Some additional results should be included for other body orientations.

Reviewer: 2

Comments to the Author(s)

Manuscript: Noncontact Respiration Monitoring Using Impulse Radio Ultrawideband (IR-UWB) Radar in Neonates

P1L37 “Neonatal movements” -> Authors probably mean “neonatal voluntary movements” instead of “neonatal chest movements”

P1 Please add information in the abstract about the respiratory status of the PICU neonates: e.g. non-mechanically assisted spontaneous breathing neonates?

P1L39 The authors describe their finding as “excellent agreement” for M0 measurements. But what do define as excellent, because the LOA are rather wide [-9.6-10.0] for a vital parameter with normal values between 30-60 breaths per minute for a neonate.

P2L33 This should be considered as part of your results: “Six neonates were eligible for the experiments among 42 full-term neonates (≥ 37 weeks of gestational age)” and is not a description of the study plan.

P2L36 Why would you exclude neonates with “dyspnea, sustained tachypnea (RR > 60 breaths/min)”? These are the interesting ones, on P2L7 the authors describe that these devices are seldom evaluated in actual clinical practices and I read this as that they are aiming to evaluate IR-UWB in a clinical setting. A device should also perform well in abnormal clinical situations.

P2L40 Did the authors preplanned how long the experiment would take per neonate? Did the authors predefine what they would consider acceptable limits of agreement?

P3L9 Is the averaging time also 10 seconds?

P3L14 I like the way the authors were able to define the extent of movement. I hope that the authors can reassure us that this method does not classify a deep breath as an M1. After all, motion of breathing is not constant, with regular changes between shallow breathing and deep breaths.

P3L37 For this type of data, with repeated measurements for every participant, it is unusual and misleading to present correlation coefficients. This very nicely explained by Bland and Altman. They also wrote an article about methods comparison studies with multiple observations per individual and how this data should be handled differently.

<http://dx.doi.org/10.1080/10543400701329422>

P3L59 I agree with the authors that this breathing trace represents matched respiratory signals. However, this does not explain why, even if only M0 measurements pairs are selected, there is still some considerable spread around the bias (represented by the rather wide LOA), especially at an average breathing rate of 40 breaths/minute. Have the authors looked into detail to these traces to understand why there’s such a large discrepancy between the devices in other measurements? Consider for example cardiac rhythm disturbances that interfere with IP signals? If the authors suspect that a large proportion of the measurement error is due to “failure” of IP measurements it would be worthwhile to look the editorial written by Myles in the Br J Anaesthesia 2007.

P4L10 I would suggest removing the correlation coefficients from the paper and also the correlation plots in Figure 4. Because correlation in this type of research is misleading.

P4L54 I am not fully convinced by the results that IR-UWB can accurately measure respiratory

rate in neonates. The bias measurement is excellent, however the LOA are rather wide also when M1 and M2 measurements are removed. A 10 breaths/minute difference can be clinically relevant. Personally, I am not so disturbed by artifacts due to voluntary movement, this indicates that the neonate is active and thus alive. I would suggest that the authors think of clinical situations in which the IR-UWB would detect breathing but the neonate is at risk of respiratory failure. E.g. upper airway obstruction, or epileptic activity. These situations should be recognized and should not produce falsely reassuring respiratory rate numbers.

Figure 5. I do not see the additive value of this figure.

Author's Response to Decision Letter for (RSOS-190149.R0)

See Appendix A.

Decision letter (RSOS-190149.R1)

24-Apr-2019

Dear Professor Park:

On behalf of the Editors, I am pleased to inform you that your Manuscript RSOS-190149.R1 entitled "Noncontact Respiration Monitoring Using Impulse Radio Ultrawideband (IR-UWB) Radar in Neonates" has been accepted for publication in Royal Society Open Science subject to minor revision in accordance with the referee suggestions. Please find the referees' comments at the end of this email.

The reviewers and Subject Editor have recommended publication, but also suggest some minor revisions to your manuscript. Therefore, I invite you to respond to the comments and revise your manuscript.

- Ethics statement

- Data accessibility

If you wish to submit your supporting data or code to Dryad (<http://datadryad.org/>), or modify your current submission to dryad, please use the following link:
<http://datadryad.org/submit?journalID=RSOS&manu=RSOS-190149.R1>

- **Competing interests**

- **Authors' contributions**

- **Acknowledgements**

- **Funding statement**

Because the schedule for publication is very tight, it is a condition of publication that you submit the revised version of your manuscript before 03-May-2019. Please note that the revision deadline will expire at 00.00am on this date. If you do not think you will be able to meet this date please let me know immediately.

When submitting your revised manuscript, you will be able to respond to the comments made by the referees and upload a file "Response to Referees" in "Section 6 - File Upload". You can use this to document any changes you make to the original manuscript. In order to expedite the

processing of the revised manuscript, please be as specific as possible in your response to the referees.

on behalf of Dr Derek Abbott (Associate Editor) and R. Kerry Rowe (Subject Editor)
openscience@royalsociety.org

Associate Editor Comments to Author (Dr Derek Abbott):

Please format your mathematics correctly. Single letter variables in the math should be in italic. All other math is not italic. So for example, "thres" should not be in italic and brackets in the math should not be in italic,

Author's Response to Decision Letter for (RSOS-190149.R1)

See Appendix B.

Decision letter (RSOS-190149.R2)

10-May-2019

Dear Professor Park,

I am pleased to inform you that your manuscript entitled "Noncontact Respiration Monitoring Using Impulse Radio Ultrawideband (IR-UWB) Radar in Neonates" is now accepted for publication in Royal Society Open Science.

on behalf of Dr Derek Abbott (Associate Editor) and R. Kerry Rowe (Subject Editor)
openscience@royalsociety.org

Appendix A

Responses to the Reviewers' Comments

We thank the editors for their constructive comments and critical review of the content. After considerable thought and discussion, we have performed additional experiments, revised the existing statistical errors, added some of our comments and edited the manuscript accordingly, as detailed below.

Reviewer: 1

1. More details about the IR-radar must be included in the text. The only reference to the radar is the company that manufacture it (Xethru X4M06; Xandar Kardian, Delaware, USA).

Thanks for your comment. The IR-UWB radar sensor is capable of detecting motions of an object using the ultra-wideband without interference from other sensors and measuring the distance to the objects. The radar module used has a center frequency of 7.29 GHz and a bandwidth of 1.5 GHz. it also uses a small power about -41.3dBm/MHz that meets Federal Communications Commision (FCC) standards (*A Lazaro et al, Sensors 2014;14:2595-2618*). In addition, it is a sensor suitable for extracting the vital signal of the patient in a non-contact method without giving any influence to the patient's body with low power. Recently, there have been researches on heart rate monitoring and arrhythmia detection using the same IR-UWB radar sensor (*Y Lee et al, Scientific reports 2018;8: 13053-13062*). And we've applied IR-UWB radar sensors to extract the respiration rate of the newborn.

As you comment, we have described more detail explanation regarding the IR-UWB radar sensors that we used.

(page 3, section 3.4, 1st paragraph)

2. In the formula of the movement $M(t)$, the difference must be defined including an absolute value of the difference $abs(m(t))$.

Thank you for taking a closer look. The difference between the frame $X(t,k)$ received from the radar and the previous frame $X(t-1,k)$, that is $X(t,k)-X(t-1,k)$ is firstly taken as an absolute value and then all the corresponding array components are added. We modified the formula correctly.

(page 3, section 3.5)

3. The signal processing described in fig.2 must be described better. For example, the

interference signal removal or vital sign extraction or noise reduction. The detection of RR must be described better. For example, if a FFT is used, what is the time-window period used, the windows used or if another algorithm is used (periodogram, ...).

In order to understand better the procedure, a new figure could be included showing a sample raw signal and the signal after each step (interference reduction, noise reduction and Fourier transform).

Thanks for your comment. In addition to radar signals received from the neonate, there are clutter signals that are received by various objects. This is because the beamforming angle of the radar antenna is wide and unwanted signals need to be removed. Background subtraction is a technique for separating the foreground from the background, where walls of static objects correspond with the background, while the neonate's chest or abdomen corresponds with the foreground. With this algorithm, background signals can be removed, and only the signal components of a respiration signals can be detected. In addition, low-pass filtering corresponding to the respiratory frequency band of the newborn has eliminated unwanted noise in the high frequency band. The time-window period is 8 seconds and the FFT Size is 4096, sampling rate is 40. I will attach an additional figure to understand the signal for each step.

In correspondence with your comment, we revised our manuscript and included a new figure (Figure 3) depicting the representative data from each step described in the block diagram. *(RS Rakibe et al, Int. J Sci Res Publ 2013;3:2250–3153)*

(page 3, section 3.4, 2nd paragraph)

4. In order to understand the effect the movement, the Fourier transform for a period with and without movement could be included. It is expected that the signal-to-noise ratios will be worst during movements.

Thanks for your comment. Since the radar basically extracts the breathing signal based on the movement of the abdomen of the newborn, the SNR of the signal drops when there is movement. Therefore, it becomes difficult to extract an accurate breathing signal while moving. I attach below breathing signals when there is movement and when there is no movement. For M0, which has little movement in the following figure, it is easy to extract the corresponding component in the frequency domain, whereas M1, M2, where movement exists, has a number of candidate points that can be the respiratory rate in the respiratory frequency band. Therefore, finding the correct respiration rate is ambiguous if movement exists.

5. The proposed method have two main drawbacks: random movements of the body and the orientation of the body. In the case of neonates the movement is small and the body falls in the illumination area of the body, therefore it is expected that is not a big problem compared with other applications, for example, a person walking. This drawback is studied in the work. The radar detects changes in the chest movement. Therefore, the signal detected in the back or lateral orientation is very small and the signal-to-noise ratio increases. In these situations, the error could be high. Some tests with different orientations of the body with respect to the radar (in lateral orientation or by back) should be included. A solution with two antennas is proposed in [18] to solve the problem. Some additional comments should be included in the discussion about this problem or potential solutions. The results given are only for the most favorable orientation. Some additional results should be included for other body orientations.

Thanks for your comment. We also believe that SNR will vary depending on the orientation of newborns. We have recently tried measurement in the supine position of newborns. Similar to those tested in the prone orientation, the figure below shows that there is good correlation between radar sensor and IP if there is no voluntary movement of the neonates in the back position. From a *signal* processing point of view, the SNR in the prone orientation may be lower than the front orientation, but there is no significant effect on detecting respiration rate. However, it is dangerous and burdensome for a newborn to experiment for a long time in a prone position. Later, studies will be conducted to detect the respiration rate in various positions in different ways. Moreover, the proposal to conduct an experiment using two or a number of radar sensors is greatly appreciated.

However, the use of multiple radars does not significantly improve accuracy or performance, so they were not applied.

Reviewer: 2

P1L37 "Neonatal movements" -> Authors probably mean "neonatal voluntary movements" instead of "neonatal chest movements"

Thanks for your comment. As your comment, "Neonatal movements" means "neonatal voluntary movement". To help understand, we have changed all the "neonatal movement" in the text to "neonatal voluntary movement".

(Summary and section 3.5)

P1 Please add information in the abstract about the respiratory status of the PICU neonates: e.g. non-mechanically assisted spontaneous breathing neonates?

Thanks for your comment. As your comment, we have added the respiratory status of the neonates in the summary of manuscript as follows; "Neonate without any respiratory support".

(Summary section)

P1L39 The authors describe their finding as "excellent agreement" for M0 measurements. But what do define as excellent, because the LOA are rather wide [-9.6–10.0] for a vital parameter with normal values between 30-60 breaths per minute for a neonate.

Thank you for the comment. We have changed the term "excellent" to "good".

(Summary and Result section)

In terms of the wide LOA issue you pointed out, we'd like to give some additional explanation. In newborn babies, periodic breathing, which is regular cycles of short apneic pauses and breaths, is a normal variation of breathing and common. Therefore, these wide LOA values can be clinically acceptable can be explained. (*JV Kraaijenga et al, Pediatric Pulm 2015;50:889-895, M Patino et al, Pediatric Anesthesia 2013;23:1166-1173, JC Cobos-Torres et al, Sensors 2018;18:4362-4375*)

P2L33 This should be considered as part of your results: "Six neonates were eligible for the experiments among 42 full-term neonates (≥ 37 weeks of gestational age)" and is not a description of the study plan.

Thank you for your comment. In accordance with your comment, we transferred description about the number of selected subjects to the result part.

(Result section)

P2L36 Why would you exclude neonates with "dyspnea, sustained tachypnea (RR > 60 breaths/min)"? These are the interesting ones, on P2L7 the authors describe that these devices are seldom evaluated in actual clinical practices and I read this as that they are aiming to evaluate IR-UWB in a clinical setting. A device should also perform well in abnormal clinical situations.

Thanks for your comment. Our study is the first pilot investigation and the aim is to study whether

the IR-UWB radar is feasible and accurate enough to be used as a continuous noncontact vital sign monitoring in the NICU environment.

In cases of sustained tachypnea, it was difficult to obtain data continuously for sufficient durations because clinical interaction and interventions between the clinical staff and the patients such as blood sampling, feeding, dressing and injections were too frequent. Although we excluded patients with sustained tachypnea, our data still have 4% of samples (N=118) with respiratory rates > 60 breaths/min since the respiratory rates of patients were fluctuating. The results (revised Figure 5 and Table 2) showed that the agreement level in patients with mean RRs ≥ 40 breaths/min was even higher than the patients with mean RRs < 40 breaths/min. In the range of respiratory rates >60 breaths/min, Bland-Altman plot showed that the mean bias level was non-significant and the width of LOAs was even narrower than those results from the whole samples (please see the figure below). Therefore, although the radar should be evaluated in the wider range of respiratory rate in future, we think that agreement between the radar and IP would be similar in the range of respiratory rates > 60 breaths/min. Moreover, we are planning for the future clinical trials in which the performance of the radar will be tested in more complicated clinical situations such as mechanical ventilation and in various patients with cardiorespiratory diseases. In accordance with your comment, we discussed about this issue more in the Discussion section.

(Discussion section, page 5, 4th paragraph).

P2L40 Did the authors preplanned how long the experiment would take per neonate? Did the authors predefine what they would consider acceptable limits of agreement?

Thanks for your comment. Before recording by the radar, we have planned to measure at least 100 min as long as the surrounding conditions of the patients were permitted. The recording was terminated while the patients met family members or visitors, or in any situations requiring a long pause of the recording. However, in short interruptions during the recording such as feeding, diaper change or injections the recording was continued. Then, we excluded the data obtained during the short interruptions from the final analysis, as we have described in the method. In accordance with your comment, the total recording times and valid recording times, in which the interrupted moments were excluded, were described in Supplementary Table 1.

Supplementary table 1. Agreement between RR_{RA} and RR_{IP} according to the individual patients

Patient	BW on recording (gram)	Age on recording (day)	Recording time (min)		N	RR	ICC	95%CI for ICC	Mean bias	95%CI for Mean bias	p-values*	95%LOA
			Total	Valid								
1	3030	2	130	93	556	42.0±6.8	0.71	0.67-0.75	0.99	0.53-1.45	<0.001	-9.83-11.81
2	2680	6	130	61	361	46.0±6.6	0.72	0.66-0.76	0.68	0.13-1.23	0.017	-9.8-11.15
3	4060	9	260	169.7	396	51.6±8.5	0.87	0.84-0.89	-0.29	-0.73-0.15	0.199	-9.12-8.53
4	2930	8	140	116.3	696	39.8±8.7	0.60	0.55-0.65	4.08	3.52-4.65	<0.001	-10.82-18.98
5	2820	12	140	98.3	588	29.6±4.0	0.59	0.53-0.64	-0.02	-0.35-0.31	0.909	-8.04-8.00
6	2940	29	140	63.3	377	50.1±7.6	0.84	0.53-0.87	-0.11	-0.56-0.34	0.629	-8.89-8.67

- derived from a one-sample t-test
- BW, body weight; RR, respiratory rate; total recording time, total time of measurement; valid recording time, time periods of M0 + M1 + M2; N of averaged samples, numbers of RR estimates averaged on a continuous basis, using 30-s sliding windows every 10 s;

P3L9 Is the averaging time also 10 seconds?

Thanks for your comment. The averaged value of respiratory rate is calculated using the values of the last 30 seconds. Samples used in actual statistics were updated every 10 seconds with the averaged values derived during the last 30 seconds.

(page 3, section 3.4, last paragraph)

P3L14 I like the way the authors were able to define the extent of movement. I hope that the authors can reassure us that this method does not classify a deep breath as an M1. After all, motion of breathing is not constant, with regular changes between shallow breathing and deep breaths.

Thanks for your comment. The logic to differentiate baby's breathing from movement as follows. When detecting patient's movements, we use autocorrelation function. Autocorrelation is the analysis of a periodic signal with noise in a given signal. When checking autocorrelation on a given respiration signal, if there is little movement, the main lobe of autocorrelation will be large by the respiratory waveform, and vice versa, if the periodic component of the waveform is eliminated by the motion due to the large movements, the main lobe will be greatly reduced. The size of the main lobe means the initial zero-crossing width of the respiratory signal through the autocorrelation function. Therefore, if the periodicity of the signals extracted from the radar is checked even with deep breathing, autocorrelation can determine that there is no movement by other bodies. Also, video recording was performed during the experiment to verify the degree of movement. The picture below is an experimental environment that combines video recording.

(Discussion section, last paragraph)

P3L37 For this type of data, with repeated measurements for every participant, it is unusual

and misleading to present correlation coefficients. This very nicely explained by Bland and Altman. They also wrote an article about methods comparison studies with multiple observations per individual and how this data should be handled differently. <http://dx.doi.org/10.1080/10543400701329422>

Thank you for the comment. As you have commented, these data are repeated measure data with unequal numbers of replicates and in pair. Because the heterogeneity that might be lurking in the data could increase the variation of the biases between the two measurements, we agree that we need to use a different analysis method. As you have recommended, we have adopted the method for the assessment of agreement with multiple observation per individual to analyze our data and revised the limits of agreement in the results. First, we performed the analysis of variance (ANOVA) test for the differences between the RR_{Radar} and RR_{IP} (Y) and subjects (X) to estimate the mean square within subjects (MS_w) and the mean square between subjects (MS_b), then calculated the variance of the difference between the RR_{Radar} and RR_{IP}, using the following formula as previously described in the article written by Bland and Altman that you have recommend.

$$\sigma_d^2 = (MS_b - MS_w) \left/ \frac{(\sum m_i)^2 - \sum m_i^2}{(n-1) \sum m_i} \right| + MS_w$$

The following R output presents the calculation of the new standard deviation for the RR in all movement levels.

```
> table(m$pt)
 1  2  3  4  5  6
556 361 396 696 588 377
----- Numbers of replicates in each subject
> (aov<-aov(data = m, dff~pt))
Call:
aov(formula = dff ~ pt, data = m)

Terms:
Sum of Squares    Df    Residuals    F value    Pr(>F)
Deg. of Freedom    5      2968
----- ANOVA (X: subjects, Y: Difference)

Residual standard error: 5.590421
Estimated effects may be unbalanced
> summary(aov)
      Df Sum Sq Mean Sq F value Pr(>F)
pt      5  8301  1660.3   53.12 <2e-16 ***
Residuals 2968  92758    31.3
---
Signif. codes:  0 '***' 0.001 '**' 0.01 '*' 0.05 '.' 0.1 ' ' 1
> aovs<-unlist(summary(aov))
> (Number.of.replicates<-(NROW(m)^2-sum((table(m$pt))^2))/((NROW(table(m$pt))-1)*NROW(m)))
[1] 489.315
----- Weighted numbers of replicates
> (MSb<-as.numeric(aovs["Mean sq1"]))
[1] 1660.291
----- Mean square between subjects
> (MSw<-as.numeric(aovs["Mean sq2"]))
[1] 31.2528
----- Mean square within subjects
> (variance<-(MSb-MSw)/Number.of.replicates+MSw)
[1] 34.58203
----- Total variance of the difference
> sqrt(variance)
[1] 5.880648
----- between the two measurement
> sd(m$dff)
[1] 5.830311
----- Standard deviation calculated using the total variance
----- Original standard deviation
```

As you can see in the analysis results, the newly calculated standard deviation is actually very similar to the original standard deviation (0.05 in difference). We think that this result occurred because there is only a small amount of heterogeneity in the data as you can see in the following figures.

We revised the results in our manuscript using the newly calculated standard deviations in accordance with your comments.

(page 3, section 3.6)

P3L59 I agree with the authors that this breathing trace represents matched respiratory signals. However, this does not explain why, even if only M0 measurements pairs are selected, there is still some considerable spread around the bias (represented by the rather wide LOA), especially at an average breathing rate of 40 breaths/minute. Have the authors looked into detail to these traces to understand why there's such a large discrepancy between the devices in other measurements? Consider for example cardiac rhythm disturbances that interfere with IP signals? If the authors suspect that a large proportion of the measurement error is due to "failure" of IP measurements it would be worthwhile to look the editorial written by Myles in the Br J Anaesthesia 2007.

First, In terms of wide LOA issue, we already described earlier.

As you pointed out, there is a short segment of discrepancies at 52 to 54 minutes as shown in Fig 3B. Interestingly, a closer look through the video camera and manual counting could verify signal perturbation in the CM in contrast to the stable radar signal. Calculation time lag could also be another cause of this discrepancies, because there is a difference between two methods to calculate the new baseline value from real-time respiratory signals whenever a big change in rate occurs. Moreover, since the time-window used to obtain respiration rate from the IR-UWB radar sensor is 8 seconds, once a movement is detected, it takes at least 8 seconds to exclude the effects of movement from the respiratory rate. Another possible explanations are an overestimation of small superficial diaphragmatic contraction or underestimation of true small breathing movement by the IP.

(Discussion section, last paragraph)

In IR-UWB radar system, interference by cardiac rhythm can be rarely occurred because RR and heartbeats are selected from the different frequency domain as follows.

P4L10 I would suggest removing the correlation coefficients from the paper and also the correlation plots in Figure 4. Because correlation in this type of research is misleading.

Thank you for the comments. As you commented, Pearson's correlation coefficients are not appropriate indexes for evaluating the levels of agreement between two raters. Therefore, we have calculated intraclass correlation coefficients and presented those for all comparisons. We agree the presentation of the Pearson's correlation r could be misleading, and we decided to remove those altogether. The scatterplot of the data was simply used to graphically present the distribution of the data. But we also agree that the drawing the pitting lines and confidence interval lines with Pearson's correlation coefficients would have misled the readers. Therefore, as you have

recommended, we decided to remove the all the correlation plot in the results and replace the one correlation plot for all data to a simple scatterplot with graphical presentation of data points from individual subjects. Furthermore, we also decided to report concordance correlation coefficient alongside with the intraclass correlation coefficients. Concordance correlation coefficient is a well-established semi-quantitative statistics to measure an agreement level between two raters for data with multiple measurements in a subject just as ours (*TS King et al, Stat Med, 2007;26:3095-3113*)

P4L54 I am not fully convinced by the results that IR-UWB can accurately measure respiratory rate in neonates. The bias measurement is excellent, however the LOA are rather wide also when M1 an M2 measurements are removed. A 10 breaths/minute difference can be clinically relevant. Personally, I am not so disturbed by artifacts due to voluntary movement, this indicates that the neonate is active and thus alive. I would suggest that the authors think of clinical situations in which the IR-UWB would detect breathing but the neonate is at risk of respiratory failure. E.g. upper airway obstruction, or epileptic activity. These situations should be recognized and should not produce falsely reassuring respiratory rate numbers.

Thank you for the comment. As we mentioned early, we used the specified Logic to differentiate between voluntary movement and breathing. Besides, the monitor can display distinctive waveforms derived from the radar. For your information, our team has recently published the paper about detection and quantification of movement of various scenarios by the IR-UWB radar (*Quantified Activity Measurement for Medical Use in Movement Disorders through IR-UWB Radar Sensor. Sensors (Basel). 2019;19(3). pii: E688. doi: 10.3390/s19030688*)

Even conventional, standard IP itself has begun to be used with similar drawbacks (TC Li, *J Perinat Med 1977;5:223-227*). Above all, the radar has the advantage of noncontact and noninvasive sensor and it can be used as a back-up cardiorespiratory monitor for a more secure monitor system, or as a main monitor for a relative healthier neonate who does not necessarily require full ECG information. As waveforms from the IR-UWB radar depict information on diaphragmatic activity and chest wall kinetics, it might address the additional value to detect and classify apnea of prematurity, wake and sleep cycles in neonates, neonatal seizures, and lung mechanics (an index of respiratory discomfort) in infants with bronchopulmonary dysplasia. This novel perspective may enable a crucial means of home monitoring for high-risk infants and early prevention of sudden infant death syndrome.

Figure 5. I do not see the additive value of this figure.

Thank you for the comment. It does seem for Figure 5 to have no additive value to the article, because it repeats the same data already shown in the other figures, including ICC and difference between the RR_{Rd} and the RR_{IP} . But actually, we have shown this figure because we would have like to show the influences of movements and RR on the bias levels and accuracy of the radar measurements. Therefore, we revised the figure so that we could present the influences of movements and RR on the bias levels and accuracy of the radar measurement more clearly, by dividing the subjects using the mean $RR \geq 40$ bpm.

Appendix B

Responses to the Reviewers' Comments

We appreciate the editor's comments and let us know that we have fixed the relevant content as follows:

- Ethics statement

- Data accessibility

<http://datadryad.org/submit?journalID=RSOS&manu=RSOS-190149.R1>

- Competing interests

- Authors' contributions

- Acknowledgements

- Funding statement

- ➔ We reviewed and revised the end statements that were mentioned at the time of each revision request. We added 'acknowledgment' to the content of the text and changed the notation of the author name. The corrected content is attached below. (red letters)

Ethics. The study protocol was approved and the study processes were monitored by the Institutional Review Board of Hanyang University Hospital, Seoul, Korea (No. 2017-09-046-002).

Data Accessibility. The dataset supporting this article has been deposited at the Dryad Digital Repository: <https://datadryad.org/review?doi=doi:10.5061/dryad.f3bn03g>

Authors' Contributions. H.K.P. and S.Ho.C. conceived the basic ideas, designed the experiments, and supervised the processes; W.H.L. and J.D.K. conducted the experiments; J.D.K., W.H.L., and Y.L. conducted data analysis and wrote the manuscript; and T.C., S.H.K., K.M.S., Y.H.L., and S.Hy.C. drafted the manuscript. All authors read and approved the final manuscript.

Competing Interests. The authors have no competing interests and no conflicts of interest to declare.

Funding. This research was supported by the Bio and Medical Technology Development Program (Next Generation Biotechnology) through the National Research Foundation of Korea (NRF) funded by the Ministry of Science, ICT and Future Planning (NRF-2017M3A9E2064735).

Acknowledgements. This content is not relevant to our study.

Associate Editor Comments to Author (Dr Derek Abbott):

Please format your mathematics correctly. Single letter variables in the math should be in italic. All other math is not italic. So for example, "thres" should not be in italic and brackets in the math should not be in italic,

- ➔ The comment mentioned was changed and it was reflected in the formula of the main manuscript. Thank you for comment. The corrected content is attached below. (red letters)

$$m(t) = \sum_{k=1}^L |X(t, k) - X(t-1, k)|$$

$$M(t) = \begin{cases} M0 & (m(t) < \text{thres1}) \\ M1 & (\text{thres1} < m(t) < \text{thres2}) \\ M2 & (m(t) > \text{thres2}) \end{cases}$$